# Inoculation with Plant Growth-Promoting Bacteria and Nitrogen Doses Improves Wheat Productivity and Nitrogen Use Efficiency

**DOI:** 10.3390/microorganisms11041046

**Published:** 2023-04-17

**Authors:** Rafaela Neris Gaspareto, Arshad Jalal, William Cesar Nishimoto Ito, Carlos Eduardo da Silva Oliveira, Cássia Maria de Paula Garcia, Eduardo Henrique Marcandalli Boleta, Poliana Aparecida Leonel Rosa, Fernando Shintate Galindo, Salatiér Buzetti, Bhim Bahadur Ghaley, Marcelo Carvalho Minhoto Teixeira Filho

**Affiliations:** 1Department of Plant Protection, Rural Engineering and Soils (DEFERS), São Paulo State University (UNESP), Ilha Solteira 15385-000, SP, Brazil; 2Center for Nuclear Energy in Agriculture (CENA), University of São Paulo (USP), Piracicaba 13416-000, SP, Brazil; 3Department of Crop Production, College of Agricultural and Technology Sciences, São Paulo State University (UNESP), Dracena 17900-000, SP, Brazil; 4Department of Plant and Environmental Sciences, Faculty of Science, University of Copenhagen, 2630 Taastrup, Denmark

**Keywords:** *Triticum aestivum* L., microorganisms, nitrogen fertilization, grain yield, cereal inoculation, nitrogen use efficiency

## Abstract

Wheat is one of the staple foods of the global population due to its adaptability to a wide range of environments. Nitrogen is one of the crucial limiting factors in wheat production and is considered a challenge to food security. Therefore, sustainable agricultural technologies such as seed inoculation with plant growth-promoting bacteria (PGPBs) can be adopted to promote biological nitrogen fixation (BNF) for higher crop productivity. In this context, the objective of the current study was to evaluate the effects of nitrogen fertilization and seed inoculations with *Azospirillum brasilense, Bacillus subtilis* and *A. brasilense* + *B. subtilis* on agronomic and yield attributes, grain yield, grain N accumulation, N use efficiency and applied N recovery in Brazilian Cerrado, which consists of gramineous woody savanna. The experiment was carried out in two cropping seasons in Rhodic Haplustox soil under a no-tillage system. The experiment was designed in a randomized complete block in a 4 × 5 factorial scheme, with four replications. The treatments consisted of four seed inoculations (control—without inoculation, inoculation with *A. brasilense, B. subtilis* and *A. brasilense* + *B. subtilis*) under five N doses (0, 40, 80, 120 and 160 kg ha^−1^, applied from urea) at the wheat tillering stage. Seed co-inoculation with *A. brasilense* + *B. subtilis* increased grain N accumulation, number of spikes m^−1^, grains spike^−1^ and grain yield of wheat in an irrigated no-tillage system of tropical savannah, regardless of the applied N doses. Nitrogen fertilization at a dose of 80 kg ha^−1^ significantly increased grain N accumulation and number of grains spikes^−1^ and nitrogen use efficiency. Recovery of applied N was increased with inoculation of *B. subtilis* and co-inoculation of *A. brasilense* + *B. subtilis* at increasing N doses. Therefore, N fertilization can be reduced by the inclusion of co-inoculation with *A. brasilense* + *B. subtilis* in the cultivation of winter wheat under a no-tillage system of Brazilian Cerrado.

## 1. Introduction

Wheat (*Triticum aestivum* L.) is one of the economically important cereals and a major staple food of the human diet. Wheat is the third most cultivated cereal worldwide, with a production of 775 million tons and a worth of more than USD 80 billion in the 2021 crop season [1]. Wheat cultivation is expanding to marginal lands to increase productivity to create food security due to the increasing global population [2]. Nitrogen (N) is one of the limiting factors that could reduce wheat production in Brazilian Cerrado, while its over-application causes leaching and off-site deposition that leads to the eutrophication of water bodies and greenhouse gases emission [3,4]. In addition, excessive application of N fertilizers could increase the cost of production, but a low-dose application can affect the performance and productivity of wheat, especially under the weathered soils of Brazilian Cerrado [5]. Therefore, proper N management is required to optimize N use efficiency and improve wheat productivity without harming the environment and to meet the increasing demand for wheat consumption [6,7].

In this context, the use of plant growth-promoting bacteria (PGPBs) is being recognized as one of the alternative techniques that could promote plant growth, N use efficiency and biological nitrogen fixation (BNF) in a sustainable way to reduce N fertilization by up to 25% of the total N applied [8,9]. Inoculation with *Azospirillum brasilense* (strains Ab-V5 and Ab-V6) has been reported as a promising inoculant for promoting plant growth and increasing wheat yield, N uptake and N use efficiency [8,10]. This inoculant has the capability to colonize the plant rhizosphere and alter root architecture by increasing root branching and volume, which could increase nutrient and water acquisition and N use efficiency [11,12]. Piccinin et al. [13] indicated that seed inoculation with *A. brasilense* can improve the agronomic performance and grain yield of wheat by allowing a 50% reduction in N fertilization.

Inoculation with *Bacillus subtilis* allows the plants to grow in abiotic extremes by stabilizing and stimulating plant growth through the solubilization of inorganic mineral phosphate and nutrient uptake [14,15]. Inoculation with *B. subtilis* also has a positive influence on the grain yield, agronomic traits and root dry mass of wheat, being considered a strategic tool in the agro-ecological production system [16,17]. Inoculation with *B. subtilis* can reduce NH_3_ volatilization up to 44% by decreasing the conversion of fertilizer N into NH_4_^+^ and increasing the nitrification process, thus increasing the N use efficiency of soil and plants [18]. Among microbial consortia, *Azospirillum* and *Bacillus* are the most reported beneficial microbes to increase the nutrient use efficiency, plant growth and productivity of different cereal crops in a sustainable manner [19,20,21].

Some studies reported that consortia of beneficial microbes are a more feasible and greener strategy to confront environmental damage due to excessive fertilizer application into agricultural soils [22]. Among microbial consortia, *A. brasilense* and *B. subtilis* are the most predominant PGPBs in different crops, soils and climatic conditions to improve nutrient acquisition along with better plant growth and yield [19,20,21]. Co-inoculation of PGPBs can stimulate different root activities that may regulate several physiological functions such as root hair elongation and meristems cell multiplication in host plants, thus leading to greater exploitation of soil for nutrient and water uptake and establishing tolerance against abiotic and biotic stresses [23,24,25].

The use of inoculants containing PGPBs is increasing day by day due to the high cost of fertilizers and increasing awareness of sustainable and less polluting agriculture. However, research on the effects of co-inoculation with *A. brasilense* and *B. subtilis* on wheat crops is still unknown and lacking. Therefore, the objective of the current study was to evaluate the combined effects of different N doses and seed inoculations with *A. brasilense* and *B. subtilis* and co-inoculation with *A. brasilense* + *B. subtilis* on agronomic parameters and production components, grain N accumulation and grain yield and N use efficiency in winter wheat in Brazilian Cerrado, which consists of gramineous woody savanna.

## 2. Materials and Methods

### 2.1. Experimental Area and Location

The experiment was carried out at the Research Station of Sao Paulo State University (UNESP), campus of Ilha Solteira, located in Selvíria–Mato Grosso do Sul–Brazil with approximate geographical coordinates of 51°22′ W, 20°22′ S and an altitude of 335 m. The specific region is called “Brazilian Cerrado”, which consists of gramineous woody savanna. The mean annual temperature and rainfall of the site are 23.5 °C and 1370 mm, respectively, with mean annual relative humidity between 70 and 80%. The climate of the region is Aw type, characterized as humid tropical with a rainy season in summer and a dry season in the winter according to the Köppen classification [26,27]. The soil is classified as Rhodic Haplustox with a clayey texture [28], being cultivated with annual crops for more than 30 years and under a no-tillage system for the last 14 years. The crop prior to wheat (the test crop of the current experiment) sowing was soybean in both the 2016 and 2017 seasons. There was an inoculation history of soybean with *Bradyrhizobium* sp. prior to the wheat experiment in both studied cropping seasons.

The daily climatic data (average, minimum and maximum temperature, rainfall and relative air humidity) during the in-field experimental duration are summarized in Appendix A.

### 2.2. Soil Analysis

The chemical attributes of the soil in the 0.0–0.2 m layer were determined before the installation of the experiment in 2016 according to the methodology proposed by Raij et al. [29] and results are summarized in Table 1.

### 2.3. Experimental Design and Treatments

The experiment in both years (2016 and 2017) was designed in a randomized complete block design with four replications in a 4 × 5 factorial scheme. The treatments consisted of four inoculations (control—without inoculation, seed inoculations with *A. brasilense* and *B. subtilis* and co-inoculation with *A. brasilense* + *B. subtilis*) and five doses of N (0, 40, 80, 120 and 160 kg ha^−1^ of N and the source of N was urea—45% of N) applied between the rows at 35 days after crop emergence (tillering or decimal growth GS21 stage). The inoculation and co-inoculation of wheat seeds were performed with *A. brasilense* strains AbV5 and AbV6 with a colony forming unit (CFU) of 2 × 10^8^ mL^−1^ and *B. subtilis* strain CCTB04 with a CFU of 1 × 10^8^ mL^−1^ at doses of 450 mL and 755 mL ha^−1^, respectively. Wheat seeds were manually inoculated by mixing with each inoculant in a separate plastic bag an hour before to plantation. Inoculations via seeds were performed as per recommendations of the inoculant-providing company (Biotrop^®^, Curitiba, Brazil). These inoculants are being commercially registered with the Ministry of Agriculture–Brazil with strains of *A. brasilense* (AzoTotal™) and *B. subtilis* (Vult™).

### 2.4. Crop Management

The experimental area has had herbicides such as glyphosate + 2,4-D (1800 + 670 g ha^−1^ of active ingredient (a.i.)) applied 15 days before wheat cultivation for the desiccation of pre-experiment emerged weeds. The seeds were chemically treated a day before plantation with a mixture of thiophanate-methyl + Pyraclostrobin (45.0 g + 5.0 g a.i.) and Fipronil (50.0 g a.i.) per 100 kg of wheat seeds. A wheat cultivar, CD1104, was sown with a mechanical seeder (Tatu Marchesan™, model PST2, Brazil) on 3 May 2016 and 10 May 2017 in a no-tillage system. Each plot consisted of 12 rows with 0.17 m spacing and 6.0 m length. Eight central rows were harvested from each plot, excluding 0.5 m from each border.

Nitrogen was applied from urea on the soil surface without incorporation into the soil on 8 June 2016 and 15 June 2017 at the tillering or decimal growth GS21 stage [30], and the area was irrigated the very next morning. Post-emergence weed management was carried out with the application of Metsulfuron Methyl (3.0 g ha^−1^ of a.i.) 20 days after emergence [31]. There was no need to control pests or diseases in wheat crops. The area was irrigated by a sprinkler system (Valley™, model 8000, Brazil) using a central pivot with an average water depth of 14 mm and irrigation shift of approximately 72 h or when necessary for the crop. Wheat harvest took place 120 and 117 days after seedling emergence, on 8 September 2016 and 12 September 2017, respectively.

### 2.5. Field Data Collection and Sample Processing

The leaf chlorophyll index was determined at the flowering stage with a ClorofiLOG^®^ model CFL-1030 device through readings of flag leaves of 10 plants per plot. Plant height (cm) was determined at physiological maturation or Zadoks stage 9 [30] of wheat by measuring the distance from the ground level to the apex of the wheat spike. Ten representative spikes of wheat were manually harvested to count the number of spikes m^−1^ and grains spike^−1^. The mass of 100 grains was determined with a precision scale of 0.01 g at 13% moisture content (wet basis) and grain yield was determined after manual collection of plants in 4 central lines of each plot. The grains were quantified after mechanical threshing and the data were transformed into kg ha^−1^ at 13% moisture content. The grain samples were placed in an airtight oven for 72 h at 65 °C to obtain dry weight. The grains were then weighed and ground in a Wiley mill for the determination of grain N concentration according to the methodology of Malavolta et al. [32]. The grain N accumulation was calculated by the following formula:Grain N accumulation (kg ha^−1^) = (N concentration × Grain yield)/1000

### 2.6. Nitrogen Use Efficiency

Nitrogen use efficiency (NUE) and recovery of applied N (RAN) were calculated by following the standard procedure of Cowden et al. [33] via the formula:NUE % = [Grain N accumulation (kg ha^−1^)] ÷ [Applied N dose (kg ha^−1^)] × 100
RAN % = [Grain + shoot N accumulation (kg ha^−1^)] ÷ [Applied N dose (kg ha^−1^)] × 100

### 2.7. Statistical Analysis

The data were initially tested with Levene’s homoscedasticity test (*p* ≤ 0.05) and then tested with Shapiro and Wilk test for normality, which showed that the data were normally distributed (W ≥ 0.90). The results were submitted to analysis of variance (F test) and Tukey’s test at 5% probability to compare the means of control treatments and inoculations with PGPB. To analyze the effect of N rates, regression equations were fitted. Statistical analyses were performed using the SISVAR program [34].

The heatmap was developed by calculating the Pearson correlation (*p* ≤ 0.05) using the corrplot package to evaluate the relationship among the evaluated attributes of wheat using R software, version 4.3.0.

## 3. Results

### 3.1. Nitrogen (N) Accumulations and Efficiencies

Grain N accumulation in wheat grains was improved with inoculations and N doses as compared to control treatments (Table 2). Co-inoculation with *A. brasilense + B. subtilis* improved wheat grain N accumulation by 31.4 and 15.8% in the 2016 and 2017 cropping seasons, respectively, which was statistically not different from the treatments with inoculation of *B. subtilis* as compared to other inoculations and control.

Nitrogen doses set a quadratic trend for grain N accumulation in both 2016 and 2017 (Figure 1A,B). Grain N accumulation was increased with increasing N fertilization up to a maximum N dose of 149 kg ha^−1^ in 2016 (Figure 1A) and up to a dose of 74.75 kg ha^−1^ of N in 2017 (Figure 1B).

Inoculation with plant growth-promoting bacteria (PGPBs) and N doses significantly influenced nitrogen use efficiency (NUE) in 2017 while the N dose effect was not significant in 2016 (Table 2). Co-inoculation with *A. brasilense* and *B. subtilis* increased NUE by 31.4% in 2016, which was statistically not different from the treatments with inoculation of *B. subtilis* as compared to those without inoculation treatments. In 2017, the treatments with inoculation of *B. subtilis* were observed with 15.8% higher NUE, which was statistically not different from the treatments with co-inoculation as compared to control treatments. In addition, N doses in 2016 were not significant for NUE; however, increasing N doses up to 83.3 kg ha^−1^ increased NUE in the second wheat cropping season (Figure 1C).

Recovery of applied nitrogen (RAN) was significantly influenced by inoculations and N doses in the 2016 and 2017 wheat cropping seasons (Table 2). Recovery of applied nitrogen was linearly increased with increasing N doses, along with inoculation of *B. subtilis* and co-inoculation of *A. brasilense* + *B. subtilis* in the 2016 cropping season (Figure 2A). The treatments without inoculation were adjusted to a quadratic trend in the 2016 cropping season, where increasing the N dose up to 76.25 kg ha^−1^ increased RAN while further increases in N doses lead to the reduction of RAN (Figure 2A). In 2017, RAN was linearly increased with increasing N doses under inoculation with *B. subtilis,* while co-inoculation and without inoculation treatments were adjusted to the quadratic equation (Figure 2B). Recovery of applied nitrogen was increased in the treatments without inoculation and with co-inoculation of *A. brasilense* + *B. subtilis* at a maximum N dose of 63.5 and 67.25 kg ha^−1^, respectively (Figure 2B). The results of both years for RAN indicated that all inoculation treatments performed better at higher N doses as compared to those without inoculation (Figure 2A,B).

### 3.2. Leaf Chlorophyll Index (LCI), Plant Height and Productive Tillers m^−1^

The interactions of inoculation with PGPBs and different N doses were not significant (*p* > 0.05) for leaf chlorophyll index (LCI), plant height (PH) and number of productive tillers m^−1^ in both wheat cropping seasons (Table 2). The effect of inoculations was significant for LCI and plant height only in the first cropping season (Table 3). Inoculation with *A. brasilense* increased LCI by 11.7% while inoculation of *B. subtilis* increased plant height by 5% as compared to without inoculation treatments in the 2016 cropping season.

In addition, productive tillers m^−1^ were increased by 33 and 18% with co-inoculation of *A. brasilense* + *B. subtilis* in 2016 and 2017, respectively, in comparison to without inoculation treatments, while the N dose effect was not significant (Table 3).

The effect of inoculations and N doses was significant (*p* > 0.01), while their interaction was not significant for the number of grains spike^−1^ in 2016 (Table 4). The treatments with co-inoculation of *A. brasilense + B. subtilis* increased the number of grains spike^−1^ by 7%, which was statistically at par with other inoculation treatments as compared to those without inoculation in the 2016 cropping season.

Nitrogen doses were set at a quadratic adjustment for the number of grains spike^−1^, where increasing N fertilization up to a maximum dose of 38.3 kg ha^−1^ of N increased the number of grains spike^−1^ in the 2016 cropping season (Figure 3).

The effect of N doses and interactions of inoculation with PGPBs and N doses on 100-grain weight and grain yield were not significant in both the 2016 and 2017 cropping seasons of wheat (Table 4). The grain yield of wheat was increased by 41 and 26% with co-inoculation of *A. brasilense* + *B. subtilis,* which was statistically not different from the treatments with inoculation of *B. subtilis* in the 2016 and 2017 cropping seasons when compared with control treatments. The lowest grain yield in both cropping seasons was noted in the treatments without inoculations (Table 4).

The current results were compared by Pearson’s correlation in 2016 (Figure 4A) and 2017 (Figure 4B) evaluating the attributes of wheat. There was a positive correlation between grain N accumulation and LCI, number of spikes m^−1^, number of grains m^−1^ and grain yield. There was a positive, but non-significant correlation between grain N accumulation and plant height, a negative correlation with N use efficiency (NUE) and recovery of applied N (RAN) and a negative and non-significant correlation with 100-grain weight. In addition, NUE has a positive correlation with RAN (Figure 4A). There was a positive correlation between grain N accumulation and LCI, number of spikes m^−1^, number of grains spikes^−1^, 100-grain weight and grain yield. In addition, there was a negative and significant correlation between all components with plant height and RAN in 2017 (Figure 4B).

## 4. Discussion

Recovery of applied N is expressed as the ability of plants to uptake N from the soil and export it to productive components for an increasing cost–benefit ratio [35]. Several factors are applied in agriculture to increase N use efficiency, such as fertilizer management and improvement in genetic traits (to increase the ability of plants to acquire more nutrients) at the crucial stage of plant development to alleviate limitations in crop growth and N demand, and adequate soil management to improve soil fertility and N availability [36]. The factors initially include more effective N application methods, site-specific N management and highly effective fertilizers (new and modified N fertilizers and inhibitors that are leading to slow/controlled release). The present results are based on inoculation and co-inoculation of *B. subtilis* and *A. brasilense*, which could fit into the aforementioned factor to improve crop growth and N use efficiency (Figure 4). It is essential to understand that several technological choices have different influences on crop yields in response to N fertilization, which might be the consequence of such practices that lead to several major benefits [36,37].

The benefits of *A. brasilense* and *B. subtilis* inoculation on root development were highlighted by greater root dry mass and root N content, which are likely to be the key mechanism to increasing NUE and RAN and lead to greater growth and grain yield of wheat. Studies reported that growth-promoting and diazotrophic bacteria have improved N acquisition by plants through biological N fixation (BNF) [5,38] and by increasing root hair growth through physiological changes in plants that have increased the production of plant growth hormones such as indole-3-acetic acid, cytokinins, gibberellins and ethylene [38,39,40], which could influence the ability of plant roots to penetrate into the soil for greater water and nutrient absorption [41].

The present study reported an increase of 14% in grain N accumulation with inoculation of *B. subtilis* and 31% with co-inoculation of *A. brasilense* + *B. subtilis* as compared to without inoculation in 2016. Meanwhile, grain N accumulation in the 2017 cropping season was increased by 14.7 and 15.8% with the same inoculations as compared to without inoculation (Table 2). The NUE was 15.8% higher with inoculation of *B. subtilis* in 2016, while in 2017, single inoculations and/or co-inoculation with *A. brasilense* and *B. subtilis* provided higher NUE at a dose of 40 kg ha^−1^ of N fertilization as compared to without inoculation. Recovery of applied nitrogen was linearly increased with inoculation of *B. subtilis* and co-inoculation of *A. brasilense* + *B. subtilis* under increasing N doses (Figure 2A,B). The possible explanation for the increase in grain N accumulation by co-inoculation with *A. brasilense* and *B. subtilis* and single inoculation with *A. brasilense* may be related to the ability of *Azospirillum* and *Bacillus* to perform biological nitrogen fixation (BNF) [38,42,43]. Although BNF is a determining factor for increasing N use efficiency and N uptake by plants, these bacteria are still functionally contributing to some other mechanisms (production of gibberellins, auxins and cytokinins) to increase plant growth [38,44]. Thus, increasing N use efficiency and the recovery of applied N in cereal crops with inoculation of *A. brasilense* [4] contributes to sustainable wheat production under reduced N fertilization [2].

Plant growth-promoting bacteria have the ability to promote plant growth through different mechanisms that could improve agronomic and yield attributes of plants, as verified in the present study in which inoculation and co-inoculation with *A. brasilense* and *B. subtilis* increased plant height, yield components and yield of wheat (Table 3 and Table 4). The difference in the performance of wheat during both growth cycles might be due to the different climatic conditions in the experimental years, with a better volume and distribution of rainfall during the 1st season of wheat (Appendix A). Previous studies reported that the interaction of microorganisms and plants activates multiple mechanisms to promote growth and improve the yield and nutritional quality of wheat, particularly with inoculation of *A. brasilense* [2,45] and *B. subtilis* [16,20]. In addition, these bacteria are used to alleviate the stressful effects of drought, thus being able to effectively contribute to the improvement of productive components and grain yield of wheat crops in tropical conditions [46,47]. As in the case of the present study, the experimental field was irrigated with supplementary irrigation but still, there is a scarcity of rainfall, with a maximum and minimum temperature above 30 and 15 °C during winter in the wheat cultivation period in Brazilian tropical savannah (Appendix A). Therefore, inoculation with *B. subtilis* and *A. brasilense* alone and/or together can contribute to food security through different defense mechanisms that increase plant tolerance to drought and harsh environmental conditions.

The isolated or combined inoculation of *A. brasilense* and *B. subtilis* proved to be effective in increasing the recovery of applied N, N use efficiency, grain N accumulation, productive components and the grain yield of winter wheat in both seasons. The N doses directly influenced the number of grains spike^−1^ and grain N accumulation, which are the correlated variables and are the main source of wheat and directly influence grain yield. Many studies carried out with irrigated wheat under similar soil and climatic conditions reported a positive correlation between leaf N concentration, yield components and grain yield with increasing N doses [48,49,50]. The highest use N efficiency and recovery of applied N were observed at the dose of 40 kg ha^−1^ of N in both years of wheat cultivation. In addition, the estimated N doses of 149 kg ha^−1^ and 75 kg ha^−1^ increased grain N accumulation by 20 and 13% in 2016 and 2017, respectively (Figure 1A,B). Nitrogen fertilization in wheat cultivation is one of the determining factors for greater productive components and better N acquisition by roots, increasing accumulation and transport of N in the stem, leaves and grains in relation to non-fertilization [8,48,51].

## 5. Conclusions

Inoculation with plant growth-promoting bacteria is considered one the most feasible, economical and sustainable strategies that could increase crop production to overcome food security challenges and reduce N fertilizer dependency. Seed inoculation with *Azospirillum brasilense* and *Bacillus subtilis* increased grain N accumulation, number of productive tillers m^−1^, grains spike^−1^ and grain yield of irrigated wheat in tropical savannah regardless of the N fertilizer doses.

The application of 40 kg ha^−1^ of N provided higher N use efficiency and applied N recovery. The increase in N fertilization positively influenced grain N accumulation and the number of grains per spike, but had no effect on grain yield.

Nitrogen fertilization in winter wheat cultivation under a no-tillage system can be reduced by the adoption of co-inoculation with *Azospirillum brasilense* and *Bacillus subtilis*.

Future studies based on long-term inoculations are encouraged to verify the beneficial effects of these bacterial strains on nitrogen fixing, soil N mobilization and mineralization and agronomic practices in changing climate on wheat productivity under field conditions. These extensive studies could be beneficial to formulating new inoculants and improving cropping systems in more profitable and eco-friendly environments with cost-effective availability to the farmer community to develop more organic and greener agriculture.

## Figures and Tables

**Figure 1 microorganisms-11-01046-f001:**
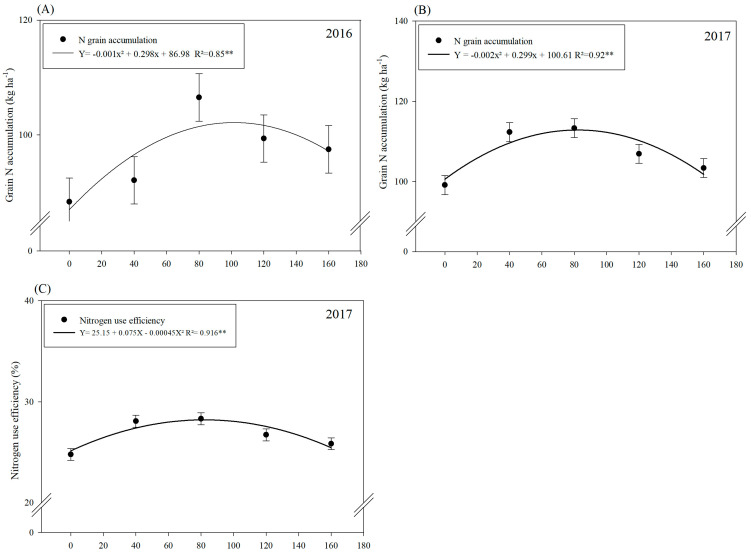
Grain N accumulation in 2016 (**A**) and 2017 (**B**); nitrogen use efficiency (NUE) in 2017 (**C**); NUE in 2016 was not significant as a function of N doses and plant growth-promoting bacterial inoculations and co-inoculation. Error bars indicate the standard deviation of the means (*n* = 4 replications). **: means it is statistically significant by Tukey’s test at probability *p* ≤ 0.01.

**Figure 2 microorganisms-11-01046-f002:**
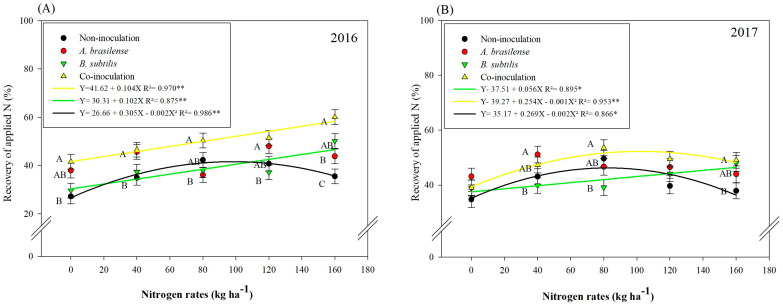
Recovery of applied N in 2016 (**A**) and 2017 (**B**) as a function of N doses and plant growth-promoting bacterial inoculations and co-inoculation. Error bars indicate standard deviation of the means (*n* = 4 replications). * and **: means it is statistically significant by Tukey’s test at probability *p* ≤ 0.05 and *p* ≤ 0.01, respectively.

**Figure 3 microorganisms-11-01046-f003:**
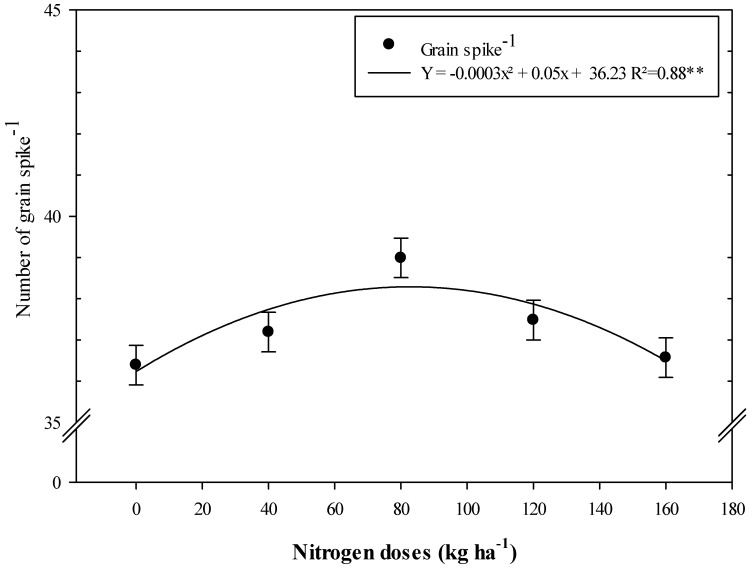
Number of grains spike^−1^ of wheat as a function of N doses in the 2016 cropping season. Error bars indicate standard deviation of the means (*n* = 4 replications). **: means it is statistically significant by Tukey’s test at probability *p* ≤ 0.01.

**Figure 4 microorganisms-11-01046-f004:**
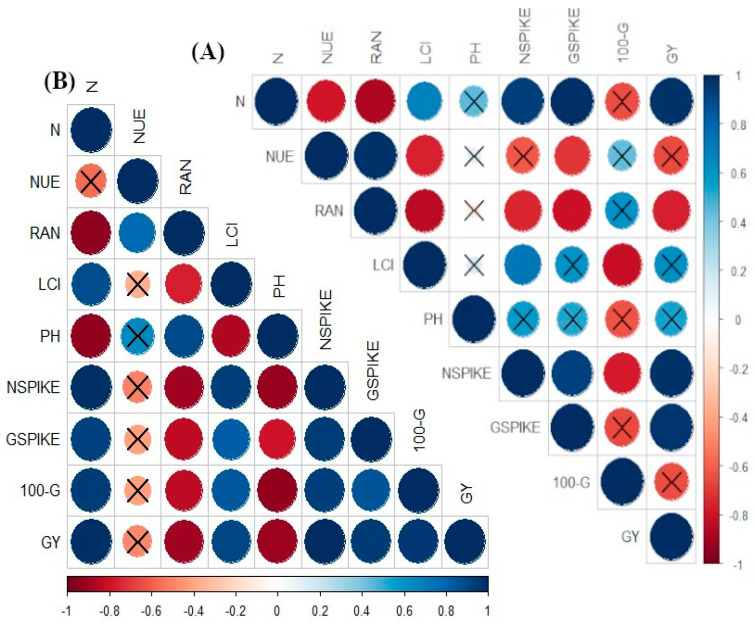
Heat-map color scale indicating Pearson’s correlation among evaluated parameters of wheat plants in response to N applications and diazotrophic bacteria inoculations in the 2016 (**A**) and 2017 (**B**) cropping seasons. X = indicates a non-significant relationship (*p* < 0.05). Abbreviations: N = grain N accumulation, NUE = N use efficiency, RA_N_ = recovery applied N, LCI = leaf chlorophyll index, PH = plant height, NSPIKE = number of spike m^−1^, GSPIKE = number of grain spike^−1^, 100-G = 100-grain weight, GY = grain yield.

**Table 1 microorganisms-11-01046-t001:** Initial soil chemical characterizations ^a^ of the experimental area in the 0.0–0.2 m layer.

Layer	P resin	S-SO_4_	OM	pH	K	Ca	Mg	H + Al
(m)	-----mg dm^−3^-----		g dm^−3^	CaCl_2_	----------------------mmol_c_ dm^−3^---------------------			
0.0–0.2	20.0	3.0	24.0	5.3	5.3	33.0	20.0	28.0
Layer	B ^b^	Cu ^c^	Fe ^c^	Mn ^c^		Zn ^c^		CEC
(m)	--------------------------mg dm^−3^--------------------------							mmol_c_ dm^−3^
0.0–0.2	0.19	3.9	21.0	63.5		1.6		86.3

^a^ Methodology proposed by Raij et al. [29], ^b^determined in hot water and ^c^ determined in DTPA (diethylenetriaminepentaacetic acid). OM: organic matter, CEC: cation exchange capacity.

**Table 2 microorganisms-11-01046-t002:** Grain N accumulation, N use efficiency and applied N recovery as a function of inoculation with plant growth-promoting bacteria and N doses in the 2016 and 2017 wheat cropping seasons.

Variables	Grain N Accumulation (kg ha^−1^)		NUE (%)		RAN (%)	
	2016	2017	2016	2017	2016	2017
**Inoculations**						
Control	85.13 ± 14.50 b	97.83 ± 7.47 c	21.28 ± 3.46 b	24.46 ± 2.47 c	36.12 ± 6.37 c	40.98 ± 5.86 c
*A. brasilense*	92.37 ± 18.92 b	104.76 ± 10.43 b	23.09 ± 5.81 b	26.19 ± 2.61 bc	42.27 ± 9.06 b	46.27 ± 4.57 ab
*B. subtilis*	97.75 ± 25.41 ab	112.25 ± 11.88 a	24.44 ± 5.58 ab	28.32 ± 3.30 a	38.46 ± 9.41 ab	41.98 ± 8.31 bc
*A. brasilense* + *B. subtilis*	111.86 13.32 a	113.30 ± 9.87 a	27.96 ± 4.17 a	28.06 ± 2.72 ab	49.97 ± 16.31 a	47.67 ± 6.89 a
**LSD**	15.46	6.71	3.86	1.98	5.11	4.81
**N Doses (kg ha^−1^)**						
0	88.37	99.18	22.90	24.79	34.09	38.94
40	92.11	112.35	23.03	28.08	41.11	45.33
80	106.54	113.32	26.64	28.33	41.65	47.24
120	99.38	106.94	24.85	26.73	44.32	44.92
160	97.49	103.39	24.37	25.84	47.35	44.69
**F-values**						
Inoculations (I)	7.49 **	11.56 **	7.49 **	11.56 **	19.77 **	6.38 **
N doses (N)	3.49 *	6.33 **	2.29 ^ns^	6.33 **	10.43 **	4.72 **
I × N	1.29 ^ns^	1.85 ^ns^	1.28 ^ns^	1.86 ^ns^	2.31 *	1.96 ^ns^
**CV(%)**	19.08	8.87	19.08	8.87	14.63	12.99

Means followed by different letters in the column are statistically significant by Tukey’s test at probability of *p* ≤ 0.05 (*) and *p* ≤ 0.01 (**). ^ns^ = not significant. NUE = nitrogen use efficiency, RAN = recovery applied N.

**Table 3 microorganisms-11-01046-t003:** Leaf chlorophyll index (LCI), plant height and number of tillers m^−1^ of wheat as a function of plant growth-promoting bacteria and N doses in the 2016 and 2017 cropping seasons.

Variables	LCI		Plant Height (cm)		Number of Tillers m^−1^	
	2016	2017	2016	2017	2016	2017
**Inoculations**						
Control	54.18 ± 5.95 b	49.77 ± 4.03 a	91.40 ± 3.16 bc	97.29 ± 5.90 a	111.80 ± 10.16 c	151.47 ± 29.39 b
*A. brasilense*	60.52 ± 5.44 a	49.54 ± 19.98 a	89.69 ± 2.81 c	96.27 ± 2.35 a	129.20 ± 21.07 b	161.07 ± 14.91 ab
*B. subtilis*	58.79 ± 5.42 ab	49.91 ± 3.65 a	96.11 ± 3.42 a	97.82 ± 2.76 a	139.55 ± 22.18 ab	170.80 ± 25.64 ab
*A. brasilense* + *B. subtilis*	57.63 ± 3.37 ab	51.26 ± 21.22 a	93.31 ± 3.59 b	96.40 ± 5.05 a	149.80 ± 14.24 a	179.87 ± 24.31 a
**LSD**	4.71	3.15	2.76	3.92	14.85	25.18
**N Doses (kg ha^−1^)**						
0	54.74	50.05	92.01	98.14	131.69	161.60
40	58.36	50.64	92.88	97.17	130.75	171.33
80	56.83	50.67	93.22	95.89	133.44	171.33
120	60.71	49.48	92.35	96.56	138.50	159.67
160	58.26	49.77	92.67	96.97	128.56	165.17
**F-values**						
Inoculations (I)	7.28 **	1.24 ^ns^	11.38 **	2.27 ^ns^	6.42 **	15.08 **
N doses (N)	0.58 ^ns^	2.01 ^ns^	1.47 ^ns^	1.92 ^ns^	0.98 ^ns^	1.18 ^ns^
I x N	1.08 ^ns^	0.78 ^ns^	0.75 ^ns^	1.15 ^ns^	1.29 ^ns^	0.88 ^ns^
**CV(%)**	8.30	7.50	3.55	4.12	13.38	15.48

Means followed by different letters in the column are statistically different from each other by Tukey’s test at a probability of *p* ≤ 0.01 (**). ^ns^ = not significant.

**Table 4 microorganisms-11-01046-t004:** Number of grains spike^−1^, 100-grain weight and grain yield of wheat as a function of inoculation with growth-promoting bacteria and N doses in 2016 and 2017.

Variables	Number of Grains Spike^−1^		100-Grain Weight (g)		Grain Yield (kg ha^−1^)	
	2016	2017	2016	2017	2016	2017
**Inoculations**						
Control	36.20 ± 3.48 b	32.53 ± 1.33 a	3.54 ± 0.28 a	3.30 ± 0.22 a	3502 ± 446 c	4066 ± 250 b
*A. brasilense*	36.90 ± 2.14 ab	33.53 ± 1.37 a	3.58 ± 0.22 a	3.44 ± 0.20 a	4037 ± 858 b	4556 ± 245 ab
*B. subtilis*	37.64 ± 2.01 ab	33.80 ± 1.86 a	3.49 ± 0.34 a	3.42 ± 0.20 a	4397 ± 978 ab	4982 ± 262 a
*A. brasilense* + *B. subtilis*	38.57 ± 3.39 a	34.33 ± 2.57 a	3.56 ± 0.22 a	3.43 ± 0.21 a	4947 ± 638 a	5152 ± 358 a
**LSD**	2.33	1.92	0.21	0.17	656.8	660.3
**N Doses (kg ha^−1^)**						
0	36.39	33.42	3.60	3.37	4162	4555
40	37.19	34.33	3.58	3.41	4086	4917
80	38.99	33.17	3.56	3.45	4475	4904
120	37.48	33.42	3.40	3.34	4262	4493
160	36.57	33.42	3.57	3.35	4122	4576
**F-values**						
Inoculations (I)	7.28 **	1.24 ^ns^	11.38 **	2.27 ^ns^	6.42 **	15.08 **
N doses (N)	5.58 **	1.07 ^ns^	1.97 ^ns^	2.02 ^ns^	1.08 ^ns^	1.77 ^ns^
I x N	1.00 ^ns^	0.88 ^ns^	0.55 ^ns^	1.07 ^ns^	1.09 ^ns^	0.38 ^ns^
**CV(%)**	7.45	5.84	7.07	6.13	18.59	14.35

Means followed by different letters in the column are statistically different from each other by Tukey’s test at a probability of *p* ≤ 0.01 (**). ns= not significant.

## Data Availability

Not applicable.

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
