# Peer review of "Inoculation with Plant Growth-Promoting Bacteria and Nitrogen Doses Improves Wheat Productivity and Nitrogen Use Efficiency"

_microorganisms, 2023, doi:10.3390/microorganisms11041046_

Round 1

Reviewer 1 Report

The manuscript deals with “Inoculation with plant growth-promoting bacteria and nitrogen 2 doses improve wheat productivity and nitrogen use efficiency”. The aim of the work (the use of plant growth promoting bacteria) is not novel at all and the paper brings low innovation to the specific sector. It has a certain value in the local context.

The experimental plan is quite traditional, but correct although some statistical analysis is missing and the correlation (if any) of the results with seasonal data (rains and temperature) is missing too. By the way, the temperature and rain data should be removed from M&M if they are not discussed or correlated with other data (EX., N uptake, and so on.).

The correct use of statistical terms and tools is required

English needs a good revision; various sentences are not clear and some of them need to be split since they are too long. Please see the enclosed PDF file for specific suggestions. Some typing/grammar errors must be corrected both in the text and in the figures.

References could be improved by citing some other recent publications on the specific field

See the enclosed PDF file with specific comments, queries and suggestions.

Author Response

Reviewer 1

The manuscript deals with “Inoculation with plant growth-promoting bacteria and nitrogen 2 doses improve wheat productivity and nitrogen use efficiency”. The aim of the work (the use of plant growth promoting bacteria) is not novel at all and the paper brings low innovation to the specific sector. It has a certain value in the local context.

The experimental plan is quite traditional, but correct although some statistical analysis is missing and the correlation (if any) of the results with seasonal data (rains and temperature) is missing too. By the way, the temperature and rain data should be removed from M&M if they are not discussed or correlated with other data (EX., N uptake, and so on.).

R: The authors thanks for all the prestegious suggestions of the reviewer. The manuscript is thoroughly revised and corrections are made as per above the comments. We hope, this version meet the expectations of the reviewer.

The correct use of statistical terms and tools is required.

R: We have added the correct information on statistics. The statistics were carried out with SISVAR program and Pearson’s correlations are calculated in R-program.

English needs a good revision; various sentences are not clear and some of them need to be split since they are too long. Please see the enclosed PDF file for specific suggestions. Some typing/grammar errors must be corrected both in the text and in the figures.

References could be improved by citing some other recent publications on the specific field

 See the enclosed PDF file with specific comments, queries and suggestions.

R: The authors sincerely grateful to the reviewer for the insight comments and suggestions. We revised the manuscript carefully and thoroughly in every aspect. Hope this version meets the expectations of the reviewer.

Thank you!

Reviewer 2 Report

To the authors

The article has an interesting point of view. However, it is not a new finding, or at least it is not well addressed in the manuscript. Unfortunately, the manuscript cannot be accepted in its present form. The article needs to improve on some points as following

Line 44 it’s -> it is (it’s, this is a no polite English form)

Line 44 Bazilian Cerrado -> What is it Cerrado? This is a Brazilian (Portuguese) word,  please define the concept, ecosystem, or agroecological condition and then the reader can understand that it is Cerrado.

 Line 46 N -> what is N? Please define it for the first time and then use it as N, not all readers are experts

Line 55 indicates the strain

Line 65 what kind of phosphate? Inorganic, accumulated, from fertilizer, or natural? Indicate

Line 69 N use efficiency “by” soil, I think this is wrong “by” rephrase it

Line 70 Azospirillum sp…. -> delete sp in both genera

Line 85 in the literature… are you sure? I recommend deleting this word

Line 86 why here use nitrogen and no N?

Line 86-87 single seed inoculation and combination of A. plus B.

Line 96 What is Aw?

Line 98 add a reference after classification. What about the inoculation history in this area? It is not cerrado, a common place for soybean inoculation. Have soybean been inoculated with Azospirillum or Bacillus in this soil? Please clarify if there is or no inoculation history in this place

Figure 1. I do not see the point of this figure in the main text; please make this point clear if not, put it as supplementary data. Also, quality needs to be improved

Line 105 please make a table about this section; it is the pH 5.3 in soil, based on what protocol?, What means DTPA?. Al is measured as what total, fragmented, partial, soluble, etc. indicates it. Only pput the reference no help the reader to understand

Line define why this is into parenthesis? Control (without inoculation), rephrase treatments. English need revised

Line 124-126. What do you mean by guarantee? It is this inoculant a commercial formulation, if it is indicate the mark country and please explain the fertilization.

Line 126 add a reference

Line 128, 131, 137, 140, 148, 149 indicate company or mark, city, and country

Line 133 – 142, indicate what type of machine, company, mark or model, country)

Line 152, 155 What do you mean by wet basis -> fresh mass or fresh weight?

Line 155 indicate dry conditions

Line 18 Figure 2 Where is nitrogen use efficiency 2016? Please improve this figure by organizing it as Left side 2016 and right side 2017 and at the same level same type of variable.

Line 159, 163, 164 add units inside the equation

Line 166 said 5% it is fine here, but actually do you mean 95% confidence?, indicate p as should be through the manuscript, unify as p

Table 1, 2, 3:  a, ab, c, etc., and then you have *, **…. Please unify it and define it based on p in the footnote, also, add SD in 2 decimals for all values. Also, it is important to unify with methods Control as control (do not use the definition of “without” here)

Line 356-364: unify the contributions by the same author in one sentence, e.g., MCMTF methodology, resources, and supervision in the same sentence (why there are 3 sentences mention only one step, simplify it, the Editor will appreciate it)

Figures quality needs to improve

Also, the discussion needs to be improved, addressing the harsh threat, problem, or difficulty for the actual issue where it is submitted. Only improving the yield is not enough.

Please revise references 1, 9, 25, 26, 27, 28, 33, 47. Also, please check if the journal’s names are long or abbreviate and unify it as indicated by the journal. Reference 1 lacks a lot of information. Is this a book, report, paper, news,etc.? Please indicate it; FAO is wide range organization. 

Author Response

Reviewer 2

The article has an interesting point of view. However, it is not a new finding, or at least it is not well addressed in the manuscript. Unfortunately, the manuscript cannot be accepted in its present form. The article needs to improve on some points as following

Line 44 it’s -> it is (it’s, this is a no polite English form)

Line 44 Bazilian Cerrado -> What is it Cerrado? This is a Brazilian (Portuguese) word,  please define the concept, ecosystem, or agroecological condition and then the reader can understand that it is Cerrado. for normality

R: The authors are thankful to the reviewer. Cerrado is a region of tropical savanna in eastern Brazil, particularly in the states of Goiás, Mato Grosso do Sul, Mato Grosso, Tocantins, Maranhão, and Minas Gerais. The main habitat types of the Cerrado consist of forest savanna, wooded savannah, park savanna and gramineous-woody savanna. Cerrado is the second largest Brazilian habitat types after the Amazonian rainforest and accounts for a full 21 percent of the country's land area. Cerrado's climate is typical of the wetter savanna regions of the world with a semi-humid tropical climate. Cerrado has two dominant seasons: rainy and dry with an annual average temperature between 22 and 27 °C and average precipitation between 80–200 cm for over 90% of the area. Therefore, we insert tropical Savannah.

 Line 46 N -> what is N? Please define it for the first time and then use it as N, not all readers are experts

R: Thanks! The N stands for nitrogen and already added to the text.

Line 55 indicates the strain

R: Thanks, Information added.

Line 65 what kind of phosphate? Inorganic, accumulated, from fertilizer, or natural? Indicate

R: Revised and indicated in the text.

Line 69 N use efficiency “by” soil, I think this is wrong “by” rephrase it

R: Revised.

Line 70 Azospirillum sp…. -> delete sp in both genera

R: Deleted as per suggestion.

Line 85 in the literature… are you sure? I recommend deleting this word

R: Deleted as per suggestion.

Line 86 why here use nitrogen and no N?

R: Replaced as suggested.

Line 86-87 single seed inoculation and combination of A. plus B.

R: Thanks. Revised.

Line 96 What is Aw?

R: The authors thanks to the reviewer. The authors already defined the Aw type climate in the paper, soon after its indication in the text. However, for further clarification for the reviewer, we are providing some more information; Aw is the second most representative climatic type on Earth (covering 11.5% of continental areas), being characterized by high temperatures throughout the year, with wet and dry seasons occurring in summer and in winter, respectively (Peel et al., 2007).

Line 98 add a reference after classification. What about the inoculation history in this area? It is not cerrado, a common place for soybean inoculation. Have soybean been inoculated with Azospirillum or Bacillus in this soil? Please clarify if there is or no inoculation history in this place

R: Thanks to the reviewer. Reference is added. The area was sown with soybean prior to wheat. Soybean was inoculated with Bradyrhizobium japonicum, which is a common practice in Brazil. The entire field of soybean was inoculated with the same inoculant and same dose.

Figure 1. I do not see the point of this figure in the main text; please make this point clear if not, put it as supplementary data. Also, quality needs to be improved

R: Figure 1 is now discussed in the discussion to better explain the difference in the results and highlighted in yellow.

Line 105 please make a table about this section; it is the pH 5.3 in soil, based on what protocol?, What means DTPA?. Al is measured as what total, fragmented, partial, soluble, etc. indicates it. Only pput the reference no help the reader to understand

R: Thanks. The data is converted into the table. All the methodologies followed for the each characteristic of the soil are being mentioned. A proper reference of Raij et al. (2001) is also being mentioned for the better understanding of the readers.

Line define why this is into parenthesis? Control (without inoculation), rephrase treatments. English need revised

R: Thanks, revised.

Line 124-126. What do you mean by guarantee? It is this inoculant a commercial formulation, if it is indicate the mark country and please explain the fertilization.

R: Thanks, the guarantee is normally used for the colony forming unit (CFU) of an inoculant, is the minimum amount of UFC that should have. The inoculants used in the current experiment are commercially recognized strains that could promote growth and productivity of cereals crops.  

Line 126 add a reference

R: Each dose of an inoculant is used as per recommendation of the inoculant providing company. The trade name of the company and inoculants are highlighted in the text. 

Line 128, 131, 137, 140, 148, 149 indicate company or mark, city, and country

Line 133 – 142, indicate what type of machine, company, mark or model, country)

R: We inserted in the manuscript this information.

Line 152, 155 What do you mean by wet basis -> fresh mass or fresh weight?

R: The grains are measured for humidity through humidity machine, where the humidity is fixed at 13%.

Line 155 indicate dry conditions

R: The grains are placed in an air-tight oven for 72 hours at 65 0C. It is a normal process for the grinding of grain material before nutritional analysis. The grains or plant material during this time is drying at uniformity and then pass through grinding process for analysis.

Line 18 Figure 2 Where is nitrogen use efficiency 2016? Please improve this figure by organizing it as Left side 2016 and right side 2017 and at the same level same type of variable.

R: Nitrogen use efficiency is clearly verified as not significant for the nitrogen doses. So it should not be mention in the figure, as we made in all the manuscript. That is the correct form. This figure is arranged according the parameters existence and occurrence year wise. We have arranged each graph in the figure by chronological order followed by year.

Line 159, 163, 164 add units inside the equation

R: Unites are added into the equations.

Line 166 said 5% it is fine here, but actually do you mean 95% confidence?, indicate as should be through the manuscript, unify as p

R: These are correct. Thanks!

Table 1, 2, 3:  a, ab, c, etc., and then you have *, **…. Please unify it and define it based on p in the footnote, also, add SD in 2 decimals for all values. Also, it is important to unify with methods Control as control (do not use the definition of “without” here)

R: Thanks, Tables are properly indicated with LSD and F values. The significance letters are allotted to the treatments with inoculations as per statistical results after analysis. The * and ** showed significance at 5% and 1%. The SD are indicated in figures interpretation for the N dose or interaction between N doses and inoculations. The without is replaced by control in all the tables.

Line 356-364: unify the contributions by the same author in one sentence, e.g., MCMTF methodology, resources, and supervision in the same sentence (why there are 3 sentences mention only one step, simplify it, the Editor will appreciate it)

R: This is according to the rules of the journal.

 Figures quality needs to improve

R: Figures resolution is improved by 300DPI.

Also, the discussion needs to be improved, addressing the harsh threat, problem, or difficulty for the actual issue where it is submitted. Only improving the yield is not enough.

R: Thank you. We revised the discussion and added the missing information for better connection.

Please revise references 1, 9, 25, 26, 27, 28, 33, 47. Also, please check if the journal’s names are long or abbreviate and unify it as indicated by the journal. Reference 1 lacks a lot of information. Is this a book, report, paper, news,etc.? Please indicate it; FAO is wide range organization. 

R; All the mentioned and other references are thoroughly checked and revised as per norms of the journal.

Thanks!

Reviewer 3 Report

Dear authors,

In this paper the authors study the effects of co-inoculation with A. brasilense and B. subtilis on wheat under five different N doses. Although there are studies that use these organisms separately, the use of both together as co-inoculum had never been tested and seems an interesting topic. The Paper is quite well written, and the data are interesting. However, I have detected a series of deficiencies in the Paper that should be corrected.

Majors:

*The introduction is very well written, citing the main literature and making a good state of art. I miss, that the authors clearly cite if Azospirillum brasilense and Bacillus subtilis can fix atmospheric nitrogen and in what condition.

*L117: “2.3. Experimental Design and Treatments” What culture medium was used to prepare the initial inoculum of A. brasilense and B. subtilis? Did it carry a source of nitrogen in its composition? Or the microorganism were growth diazotrophicly, Please indicate.

*L126:” at doses of 450 ml and 755 ml ha-1 respectively” Based on what were those volumes for inoculation chosen? Have been based on any specific reference? Please indicate it.

*Table 1, table 2, table 3: I do not understand these tables, the different N doses,  to which inoculation conditions were applied?  Why is it not indicated? To which N doses the different inoculations were carried out, Why is it not indicated?

*Fig. 2: Why is it not shown the inoculation effect on the N grain accumulation Fig 2 A and B? and in the nitrogen use efficiency Fig 2C? Do not have those data? If you have that data, you should show them and discuss them.

* The conclusions must be remade; half of the first paragraph is not a conclusion. Better include the final part apart as a future perspective section.

*L290:” The NUE was 158% higher with inoculation of B. subtilis” Review this data, I think it's wrong

*L294:” an increase of 65 and 147%”. Please, discuss what this difference could be due.

*L299: “carrying similar fix and nif genes that are contributing to BNF and enhancing” What do you mean by this? How that similar fix and nif Genes? similar in what? What are those nif genes and which enhance the reductase nitrate?

Minors:

*L141: “here was no need to control pests or diseases in wheat crops” why? Why not, Please explain.

* In the legend of Fig. 1 indicate what HR means (relative humidity).

*Table 1, 2, 3: CV(% . Typo

* Fig. 3 and Fig 4 A has little sharpness

*L347: “influenced n N grains” Typo

*L284: “plant growth hormones” Please indicate what hormones are

Author Response

Reviewer 3

In this paper the authors study the effects of co-inoculation with A. brasilense and B. subtilis on wheat under five different N doses. Although there are studies that use these organisms separately, the use of both together as co-inoculum had never been tested and seems an interesting topic. The Paper is quite well written, and the data are interesting. However, I have detected a series of deficiencies in the Paper that should be corrected.

Majors:

*The introduction is very well written, citing the main literature and making a good state of art. I miss, that the authors clearly cite if Azospirillum brasilense and Bacillus subtilis can fix atmospheric nitrogen and in what condition.

R: The author thanks to the reviewer. We have provided information about biological nitrogen fixation in our introduction, which is the major mechanisms of these inoculants for promoting plant growth, yield and nutrients use efficiency.

*L117: “2.3. Experimental Design and Treatments” What culture medium was used to prepare the initial inoculum of A. brasilense and B. subtilis? Did it carry a source of nitrogen in its composition? Or the microorganism were growth diazotrophicly, Please indicate.

R: These are commercial strains. We had directly bought these inoculant from the bio-inoculants company. These inoculants are being test and verified their activities of growth promotion and increasing bioavailability of nutrients and biological nitrogen fixation in several studies. The genome sequences of A. brasilense described that both strains Ab-V5 and Ab-V6 carried fix and nif genes that help in the promotion of nutrient transportation, biological nitrogen fixation, production of phytohormones and invigorate tolerance against abiotic stresses (Fukami et al. 2018a, b; Hungria et al. 2018). Bacillus subtilis is carrying non-ribosomal peptide synthetases and beta-glucanase to prevent phyto-pathogen infestation, help in bioremediation of heavy metal and Zn transporter (zntR) that promote plant growth (Chaoprasid et al., 2015; Rekha et al., 2017; Muñoz-Moreno et al., 2018).

*L126:” at doses of 450 ml and 755 ml ha-1 respectively” Based on what were those volumes for inoculation chosen? Have been based on any specific reference? Please indicate it.

R: We have been tested these inoculants in so many studies and found these best for the inoculation of wheat seeds. The current experiment is not only related to bacterial inoculation but also inoculants combination with nitrogen doses. This combination itself is a unique combination and no studies are being conducted so far in Brazilian tropical soil. The doses of inoculants are being selected as per classification of the inoculant providing company and some preliminary experiments (not published yet). 

*Table 1, table 2, table 3: I do not understand these tables, the different N doses,  to which inoculation conditions were applied?  Why is it not indicated? To which N doses the different inoculations were carried out, Why is it not indicated?

R: The N doses are applied to all the inoculation in proper treatments combination. However, where there is significant interaction of N doses and inoculations. We have differentiated by the significance letters while where the interaction is not significant, it showed in the figures in combination.

*Fig. 2: Why is it not shown the inoculation effect on the N grain accumulation Fig 2 A and B? and in the nitrogen use efficiency Fig 2C? Do not have those data? If you have that data, you should show them and discuss them.

R: Thanks. First of all the experiment was designed in a randomized complete block in a 4 x 5 factorial scheme. The treatments combination were consisted of four seed inoculations (control-without inoculation, inoculation with A. brasilense, B. subtilis and A. brasilense + B. subtilis) under five N doses (0, 40, 80, 120 and 160 kg ha-1, applied from urea). Therefore, we have shown in the tabulated interpretation our data that the interactions of inoculations and N doses are not significant. The only effect of N doses are significant for the attributes shown in the Figure 2. We have already interpreted the data in the Table 2 and so on for the rest of the figures.

* The conclusions must be remade; half of the first paragraph is not a conclusion. Better include the final part apart as a future perspective section.

R: Thanks. Conclusions are revised and future prospective are being added.

*L290:” The NUE was 158% higher with inoculation of B. subtilis” Review this data, I think it's wrong

R: Revised and corrected.

*L294:” an increase of 65 and 147%”. Please, discuss what this difference could be due.

R: There was some typographical mistakes. The results interpretations are revised and addressed the raised issue.

*L299: “carrying similar fix and nif genes that are contributing to BNF and enhancing” What do you mean by this? How that similar fix and nif Genes? similar in what? What are those nif genes and which enhance the reductase nitrate?

R: This is a part of the discussion and not of the original results. The authors are trying provide a dynamic and mechanistic literature review. This explanation is based on the gene sequencing of the inoculants used in the current study. However, we deleted this part as it was producing doubts. 

Minors:

*L141: “here was no need to control pests or diseases in wheat crops” why? Why not, Please explain.

R: There are several reasons that didn’t allow us to use any pesticides etc. 1). We have chemically treated wheat seeds with insecticides and fungicides before cultivation for avoiding soil pathogens attack. 2). Wheat are sowing in the dry season in this region, where there is low humidity in the air and less or no chance of fungus attack. 3). We have a proper irrigation at a required time. 4). Wheat is not a major crop in tropical savannah of Brazil. Hence, there was no other cultivated field in the time of current experiment that also reduced chances of diseases and pests transfer into the current experiment.

* In the legend of Fig. 1 indicate what HR means (relative humidity).

R: Yes, it relative air humidity that we already mentioned in the full form.

*Table 1, 2, 3: CV(% . Typo

R: Corrected.

* Fig. 3 and Fig 4 A has little sharpness

R: The figures resolution is increased to 300 dpi.

*L347: “influenced n N grains” Typo

R: Corrected.

*L284: “plant growth hormones” Please indicate what hormones are

R: Thanks. Information are added to the text and highlighted for your kind consideration.

Thanks!

Round 2

Reviewer 1 Report

 Although the authors improved the manuscript with various suggestion from the different reviewers the main point discussed by this reviewer had not been neither addressed nor commented (the authors just addressed the easy points):

11)      They still show in M&M some information regarding the rain during the experimentation years. As discussed before they must be removed or discussed possibly performing the relative analysis to correlate the precipitations with their results

22)      The tables were not corrected as suggested: statistical analysis is not given for the treatments relative to different N treatments. Moreover, SD has not been added to the data.

Please look at the PDF enclosed during the first revision

This is mandatory for manuscript acceptance

Author Response

Responses to Reviewer’s Comments and Suggestions

 Although the authors improved the manuscript with various suggestion from the different reviewers the main point discussed by this reviewer had not been neither addressed nor commented (the authors just addressed the easy points):

R: The authors thanks to the reviewer. We really appreciate the vision of the reviewer. We addressed most of the suggestions and comments of the reviewer previously. We hope that we have answered every inquiry to your satisfaction and also hope that you will find this version of publishable quality. Hope, this version has met the expectations of the reviewer.

They still show in M&M some information regarding the rain during the experimentation years. As discussed before they must be removed or discussed possibly performing the relative analysis to correlate the precipitations with their results

R: Thanks to the reviewer. Yes, we agree with the reviewer that we didn’t do relative analysis to correlate the precipitation data to the results of the current manuscript. However, we didn’t delete this figure because we already discussed Figure 1 in the discussion of the manuscript. Figure 1 showed a distinct difference in both cropping seasons, which provide an opportunity to the authors explain difference in the results of first and second cropping season. This figure is cited in the discussion to support our results.   

The tables were not corrected as suggested: statistical analysis is not given for the treatments relative to different N treatments. Moreover, SD has not been added to the data.

R: Thanks to the reviewer. The tables are being revised. The statistics are being carried out with SISVAR program and Pearson’s correlations between the evaluated attributes of the current study are calculated in R-program. Statistics showed that whenever, there are quantitative factors with more than four doses should be performed with regression analysis. While, the difference in the treatments with regression analysis should not be placed with significance differences. In the current case, Nitrogen has five doses and evaluated with regression analysis. That’s why the authors followed international statistics rules and didn’t indicated the treatments under N doses with significance letters. In addition, we calculated and inserted the SD values for each evaluation according to the isolated treatments of inoculation with growth-promoting bacteria, in which, as it is a qualitative factor, the Tukey test was applied to compare the means in the tables.

Please look at the PDF enclosed during the first revision.

R: The authors thanks to the reviewer. We looked into the PDF of the first revision of the reviewer and answered each and every point in the first revision of the manuscript. Hope this version met the expectations of the reviewer.

This is mandatory for manuscript acceptance.

R: Our most sincere gratitude to the reviewers who took time from his busy schedules to help us and make this manuscript a better paper.

Reviewer 2 Report

To the authors

Thank you for revising your manuscript.

I recommend the authors ask an English native or colleague with English knowledge and check the manuscript. It needs to improve. 

Please unify the decimals through the manuscript you cannot use 0.25m,  0.5 m, and the 6 m in the same paragraph, it does not look good, please unify them as one or 2 decimals e.g. 0.25 m, 0.5 m, 6.0 m or 45.0 g + 5.0 g, 670.0g

Also, unify hours as h. e.g. lines 148, 161, etc.

Line 44 it’s -> its

Line -64 acidic and saline soils, and pesticides exposure etc. -> acidic and saline soils, pesticides exposure and others (etc. is not exactly polite).

Line 65 mineral phosphate -> what do you think about put as “inorganic mineral phosphate”?

Line 69 NH4-N -> what means-N?

Line 90 in the tropical savannah region of Brazil -> in the tropical savannah areas or called Cerrado region of Brazil (is it like this?).

Line 95 before the mean annual, please add this description as The specific region is called “Brazilian Cerrado”, which consist of? Please pick the corresponding or ones to the study area -> forest savanna, wooded savannah, park savanna, and gramineous-woody savanna. And then you can continue with The mean annual…

 Line 98-99 Thank you for the explanation about Aw. Please add a reference (Peel et al., 2007) after (Aw type or next to reference 26).

Line 110: 0-0.20-> 0.0 - 0.20m

Table 1 Please improve the table e.g., decimals should be the same .0 or .00 (1 or 2 decimals, those values that do not have decimals put .0). what means b, check the footnote

Also, it is confusing, how about this example

Minerals

units

Layer 1

Layer 2

Presin

mg dm-3

xx

xx

OM

g dm-3

xx

xx

Line 115 reference 2001 change to 28?

Line 132 Has the B. subtilis been used before in soybean crops in combination with Bradyrhizobium in this field? If my assumption is correct please add one sentence that there is an inoculation history with XX in soybean previously this experimental data.

Line 134-145 what does mean a.i? where are these compounds' company, city, and country used it? 

Line 154 add a reference at the end of the sentence for Zadoks stage 9.  

Line 161 for drying -> for dried mass or dried weight.

Line 165 thank you for adding units but still, please add units to the equation

Table 1, 2, 3, 4 Please check again carefully and unify the decimals to 2. For example, in Table 2 in 0 rows 2016 is it 22.90?. Change = to equal to. Also, add the SD for each value Tukey’s test indicates SD for each value, for example, 85.13±15.46 b (15.46 it is not for each value). This is field data, it is important to indicate the value and variabilities for each variable.  Tukey -> Tukey’s. Also, it seems repetitive but it is better if you indicate the p ≤ 0.05 and p ≤ 0.01

Figure 2 I got your explanation, actually, NS data is still data. Anyway, at least, please move the fig. D to down before Fig. E keep an empty space after Fig. C. You should keep the logical and order for readers easy to understand and make comparisons and also the style is better.  This can figure can easily be modified in pptx.

Line 356-364: unify the contributions by the same author in one sentence, e.g., MCMTF methodology, resources, and supervision in the same sentence (why there are 3 sentences mention only one step, simplify it) this is a rule for an indication but it is not a strict paragraph, you must avoid redundancy in the manuscript. It decreases the quality and value of the article.

Figure 3 needs to improve it is blurry

Discussion lines 341-344 delete figure words (this is a discussion section, no need for that). Same for tables from lines 356-363

Line 374 mentions harsh conditions is not mean, this is a discussion there is a wide range of harsh conditions, please indicate the respective one, for example, savanna lack rainfall as you mentioned drug harsh conditions but how about high moisture in the rainy season and the acidic soil conditions and p accumulation from fertilizer, also how about Al toxicity? Please improve the discussion for savanna harsh conditions. I can only see an improvement in wheat yield in the savanna. 

Reference italics in journal names e.g. Ref. 5

Author Response

Responses to the reviewer’s Comments and Suggestions

Thank you for revising your manuscript.

I recommend the authors ask an English native or colleague with English knowledge and check the manuscript. It needs to improve. 

R: The authors thanks to the reviewer. We revised the manuscript for English language and all other suggestions and comments of the reviewer. We hope that we have answered every inquiry to your satisfaction and also hope that you will find this version of publishable quality. Hope, this version has met the expectations of the reviewer.

Please unify the decimals through the manuscript you cannot use 0.25m,  0.5 m, and the 6 m in the same paragraph, it does not look good, please unify them as one or 2 decimals e.g. 0.25 m, 0.5 m, 6.0 m or 45.0 g + 5.0 g, 670.0g

Also, unify hours as h. e.g. lines 148, 161, etc.

R: Thanks, all the pointed and rest of the values in the manuscript are revised and unified as suggested.

Line 44 it’s -> its

R: Corrected. Thanks.

Line -64 acidic and saline soils, and pesticides exposure etc. -> acidic and saline soils, pesticides exposure and others (etc. is not exactly polite).

R: Removed. Thanks for indicating.

Line 65 mineral phosphate -> what do you think about put as “inorganic mineral phosphate”?

R: Corrected as per suggestion of the respective reviewer.

Line 69 NH4-N -> what means-N?

R: Here the authors mean that N was converted into NH4+, also corrected in the main text. Thanks

Line 90 in the tropical savannah region of Brazil -> in the tropical savannah areas or called Cerrado region of Brazil (is it like this?).

R: The authors like to keep Brazilian Cerrado, which is the most correct term for experimental field. To standardize, the authors keep Brazilian Cerrado in the entire manuscript. 

Line 95 before the mean annual, please add this description as The specific region is called “Brazilian Cerrado”, which consist of? Please pick the corresponding or ones to the study area -> forest savanna, wooded savannah, park savanna, and gramineous-woody savanna. And then you can continue with The mean annual…

R: Thanks, The authors insert this information in the material and method as per suggestion of the reviewer.

 Line 98-99 Thank you for the explanation about Aw. Please add a reference (Peel et al., 2007) after (Aw type or next to reference 26).

R: Thanks, reference inserted as per suggestion.

Line 110: 0-0.20-> 0.0 - 0.20m

R: Thanks for highlighting. It was typo mistakes, which is now corrected.

Table 1 Please improve the table e.g., decimals should be the same .0 or .00 (1 or 2 decimals, those values that do not have decimals put .0). what means b, check the footnote

Also, it is confusing, how about this example

Minerals

units

Layer 1

Layer 2

Presin

mg dm-3

Xx

Xx

OM

g dm-3

Xx

Xx

 R: The authors thanks to the reviewer. The decimals are unified all over the text and tables. We like this table however, we have done soil analysis only in soil layer (0.0 - 0.2 m) before the installation of the experiment. There was a typo mistake, which is addressed and corrected. Hope this, time confusion should be clear and the table will be easy for the reader in the present form.

Line 115 reference 2001 change to 28?

R: Thanks, corrected.

Line 132 Has the B. subtilis been used before in soybean crops in combination with Bradyrhizobium in this field? If my assumption is correct please add one sentence that there is an inoculation history with XX in soybean previously this experimental data.

R: The area was sown with soybean prior to wheat. Soybean was inoculated with Bradyrhizobium japonicum, which is a common practice in Brazil. The entire field of soybean, planted before the wheat plantation, was inoculated with the same inoculant and same dose. This information is briefly added to the main text of the manuscript in materials and method section.

Line 134-145 what does mean a.i? where are these compounds' company, city, and country used it? 

R: Thanks, a.i. means active ingredient. As its full form was mentioned before that’s why it was making some confusion however, now its corrected in the manuscript.

Line 154 add a reference at the end of the sentence for Zadoks stage 9.  

R: Thanks, added.

Line 161 for drying -> for dried mass or dried weight.

R: Thanks, corrected.

Line 165 thank you for adding units but still, please add units to the equation

R: Thanks, added.

Table 1, 2, 3, 4 Please check again carefully and unify the decimals to 2. For example, in Table 2 in 0 rows 2016 is it 22.90?. Change = to equal to. Also, add the SD for each value Tukey’s test indicates SD for each value, for example, 85.13±15.46 b (15.46 it is not for each value). This is field data, it is important to indicate the value and variabilities for each variable.  Tukey -> Tukey’s. Also, it seems repetitive but it is better if you indicate the p ≤ 0.05 and p ≤ 0.01

R: Thanks, the tables are revised and unified with decimals. SD are added for the Tukey’s test as suggested. The changes suggested are also made.

Figure 2 I got your explanation, actually, NS data is still data. Anyway, at least, please move the fig. D to down before Fig. E keep an empty space after Fig. C. You should keep the logical and order for readers easy to understand and make comparisons and also the style is better.  This can figure can easily be modified in pptx.

R: The figure 2 is divided into two figures. I hope this version has met the expectations of the reviewer.

Line 356-364: unify the contributions by the same author in one sentence, e.g., MCMTF methodology, resources, and supervision in the same sentence (why there are 3 sentences mention only one step, simplify it) this is a rule for an indication but it is not a strict paragraph, you must avoid redundancy in the manuscript. It decreases the quality and value of the article.

R: Thanks, revised as suggested.

Figure 3 needs to improve it is blurry

R: Thanks, the figure is added in best quality this time.

Discussion lines 341-344 delete figure words (this is a discussion section, no need for that). Same for tables from lines 356-363

R: Thanks, deleted as per suggestion.

Line 374 mentions harsh conditions is not mean, this is a discussion there is a wide range of harsh conditions, please indicate the respective one, for example, savanna lack rainfall as you mentioned drug harsh conditions but how about high moisture in the rainy season and the acidic soil conditions and p accumulation from fertilizer, also how about Al toxicity? Please improve the discussion for savanna harsh conditions. I can only see an improvement in wheat yield in the savanna. 

R: The harsh conditions are mentioned in the sense, this region was receiving less precipitation in the last three years as shown in the figure. Moreover, the soil characterization is already described in the table 1. We had already discussed the role of PGPBS and their mechanisms dealing with all the harsh conditions. Productivity was the main focus of this study that’s why the authors emphasized on it. Hope the reviewer should understand our point of view. So many thanks.

Reference italics in journal names e.g. Ref. 5

R: Thanks, corrected.

Reviewer 3 Report

Dear Authors,

I believe that the authors have responded favorably to most of my suggestions, and I accept the paper in its latest version.

Author Response

Thanks!